# Organization of the *Drosophila* larval visual circuit

Ivan Larderet[1†], Pauline MJ Fritsch[1†], Nanae Gendre[1], G Larisa Neagu-Maier[1], Richard D Fetter[2], Casey M Schneider-Mizell[2], James W Truman[2], Marta Zlatic[2], Albert Cardona[2], Simon G Sprecher[1]*

[1]Department of Biology, University of Fribourg, Fribourg, Switzerland; [2]Janelia Research Campus, Howard Hughes Medical Institute, Ashburn, United States

**Abstract** Visual systems transduce, process and transmit light-dependent environmental cues. Computation of visual features depends on photoreceptor neuron types (PR) present, organization of the eye and wiring of the underlying neural circuit. Here, we describe the circuit architecture of the visual system of *Drosophila* larvae by mapping the synaptic wiring diagram and neurotransmitters. By contacting different targets, the two larval PR-subtypes create two converging pathways potentially underlying the computation of ambient light intensity and temporal light changes already within this first visual processing center. Locally processed visual information then signals via dedicated projection interneurons to higher brain areas including the lateral horn and mushroom body. The stratified structure of the larval optic neuropil (LON) suggests common organizational principles with the adult fly and vertebrate visual systems. The complete synaptic wiring diagram of the LON paves the way to understanding how circuits with reduced numerical complexity control wide ranges of behaviors.

DOI: https://doi.org/10.7554/eLife.28387.001

**\*For correspondence:**
simon.sprecher@gmail.com

[†]These authors contributed equally to this work

**Competing interests:** The authors declare that no competing interests exist.

## Introduction

Light-dependent cues from the surrounding world are perceived by specialized photoreceptor neurons (PRs) in the eye. Insect compound eyes are elaborate systems capable of mediating flight in a rapidly changing 3D environment. In contrast, larval stages present much simpler visual organs, which combined with their tractability, make them great models to link neural circuit processing and behavior (*Kane et al., 2013*; *Randel et al., 2014*; *2015*; *Gepner et al., 2015*). Larvae of the fruit fly *Drosophila melanogaster* employ their visual system for a range of diverse behaviors including navigation, entrainment of circadian rhythms, formation of associative memories and may respond to the presence of other larvae (*Kane et al., 2013*; *Humberg and Sprecher, 2017*; *Slepian et al., 2015*; *Justice et al., 2012*; *Yamanaka et al., 2013*; *von Essen et al., 2011*; *Gong, 2009*; *Mazzoni et al., 2005*; *Gerber et al., 2004*; *Sawin-McCormack et al., 1995*). The simple eyes of the larva (also termed Bolwig Organ, BO) consist of only about 12 PRs each and yet drive a wide range of behaviors, raising questions on the organizational logic of the underlying visual circuit. Spectral sensitivity of PRs is defined by the *Rhodopsin* gene they express. Larval eyes contain two PR-types, either expressing the blue-tuned *Rhodopsin5* (Rh5) or the green-tuned *Rhodopsin6* (*Rh6*) (*Malpel et al., 2002*; *Hassan et al., 2005*; *Rodriguez Moncalvo and Campos, 2005*; *Sprecher et al., 2007*). Interestingly, for rapid navigation away from light exposure only the Rh5-PRs seem essential, whereas to entrain the molecular clock either PR subtype suffices (*Keene et al., 2011*). In the past, several neurons of the larval visual neural circuit have been identified but the logic of circuit wiring as well as the precise numbers of neurons involved in its first order visual processing center remain unknown.

Larval PRs project their axons in a joint nerve (Bolwig nerve) terminating in a small neuropil domain termed the larval optic neuropil (LON). Previous studies identified eleven neurons

innervating the LON in each brain hemisphere. This includes four lateral neurons (LaNs) expressing the pigment dispersing factor (Pdf) neuropeptide (Pdf-LaNs) and a fifth non-Pdf-expressing LaN (5th-LaN), all being part of the clock circuit (*Kaneko et al., 1997*), as well as a serotonergic neuron and three optic lobe pioneer cells (OLPs) (*Helfrich-Förster, 1997*; *Rodriguez Moncalvo and Campos, 2005*; *2009*; *Tix et al., 1989*). Recently, two unpaired median octopaminergic/tyraminergic neurons were described to extend neurites into these neuropils (*Selcho et al., 2014*). It remained unknown, however, whether these previously identified neurons constitute the entire neuronal components of the LON and how visual neuronal components connect to each other to form a functional network. In a recent study we started to investigate the anatomy of the LON using serial-section transmission electron microscopy (ssTEM) showing that PRs' axons form large globular boutons with polyadic synapses and that the OLPs were parts of their direct targets (*Sprecher et al., 2011*).

Here, we mapped the synaptic wiring diagram of the LON by reconstructing all its innervating neurons from a new ssTEM volume of a whole first instar larval central nervous system (*Ohyama et al., 2015*). We characterized and quantified the connectivity of all previously described LON-associated neurons and identified new components in this circuit. We found per hemisphere eleven second order interneurons, two third-order interneurons and one serotonergic neuron innervating the LON, plus two unpaired octopaminergic/tyraminergic neurons contacting both hemispheres. We highlighted the separation of light signal flow at the first synapse level as the two PR subtypes connect onto distinct subsets of interneurons. Network analysis suggests that the resulting circuits may encode both ambient light intensity information and information about the changes of light intensity. Comparison with the visual circuit of the adult fruit fly highlights common principles in organization of visual information processing, for example the stratification of PRs inputs as well as the existence of distinct photoreceptor pathways that are involved in detecting temporal light cues. Furthermore, the comparison with the olfactory wiring diagram (*Berck et al., 2016*) highlights common strategies for early sensory information processing, relay to higher order areas such as the mushroom bodies for associative memory, and control from the central brain. By mapping the connectivity of visual circuits and analyzing its architecture we have the opportunity to study the circuit structure-function relationship and advance our understanding of how neural circuits govern behavior.

## Results

### Neurons of the larval visual circuit

Axonal projections of larval photoreceptor neurons (PRs) enter the brain lobes ventro-laterally via the Bolwig nerve and terminate in a small neuropil domain, termed larval optic neuropil (LON; *Figure 1A,B*, *Figure 1—video 1*). Visual interneurons innervate the LON from the central brain through the central optic tract (*Sprecher et al., 2011*). We reconstructed the axon terminals of all PRs and all their synaptic partners, as well as additional LON-innervating neurons that do not form synapses with PRs, from a serial-section transmission electron microscopy (ssTEM) volume spanning the complete central nervous system of a *Drosophila* first instar larva (using the Collaborative Annotation Toolkit for Massive Amounts of Image Data (CATMAID), *Saalfeld et al. (2009)*; *Ohyama et al. (2015)*; *Schneider-Mizell et al., 2016*). In this way, we identified the complete repertoire of LON neurons and mapped the wiring diagram of the left and right LONs.

We define five neuron types (*Figure 1A,B*): first, sensory neurons (photoreceptor neurons) that innervate the LON; second, visual local interneurons (VLNs) that do not extend neurites beyond the LON; third, visual projection interneurons (VPNs) that relay signals from the LON to distinct higher brain areas; fourth, third-order interneurons in the LON that do not receive direct input from the PRs; and fifth, modulatory aminergic feedback neurons projecting from the central brain.

Two interneurons belonging to the previously described optic lobe pioneer cells (OLPs, *Tix et al., 1989*) are VLNs, which we therefore named local-OLPs (lOLPs). Their arbors are fully contained within the LON and they present a distinct axon and dendrite (*Figure 1C,D*), comparable to glutamatergic inhibitory neurons of the larval antennal lobe (*Berck et al., 2016*). We found that one lOLP is cholinergic (cha-lOLP) while the other is glutamatergic (glu-lOLP), in agreement with previous studies (*Figure 1—figure supplement 1*; *Yasuyama et al., 1995*; *Daniels et al., 2008*).

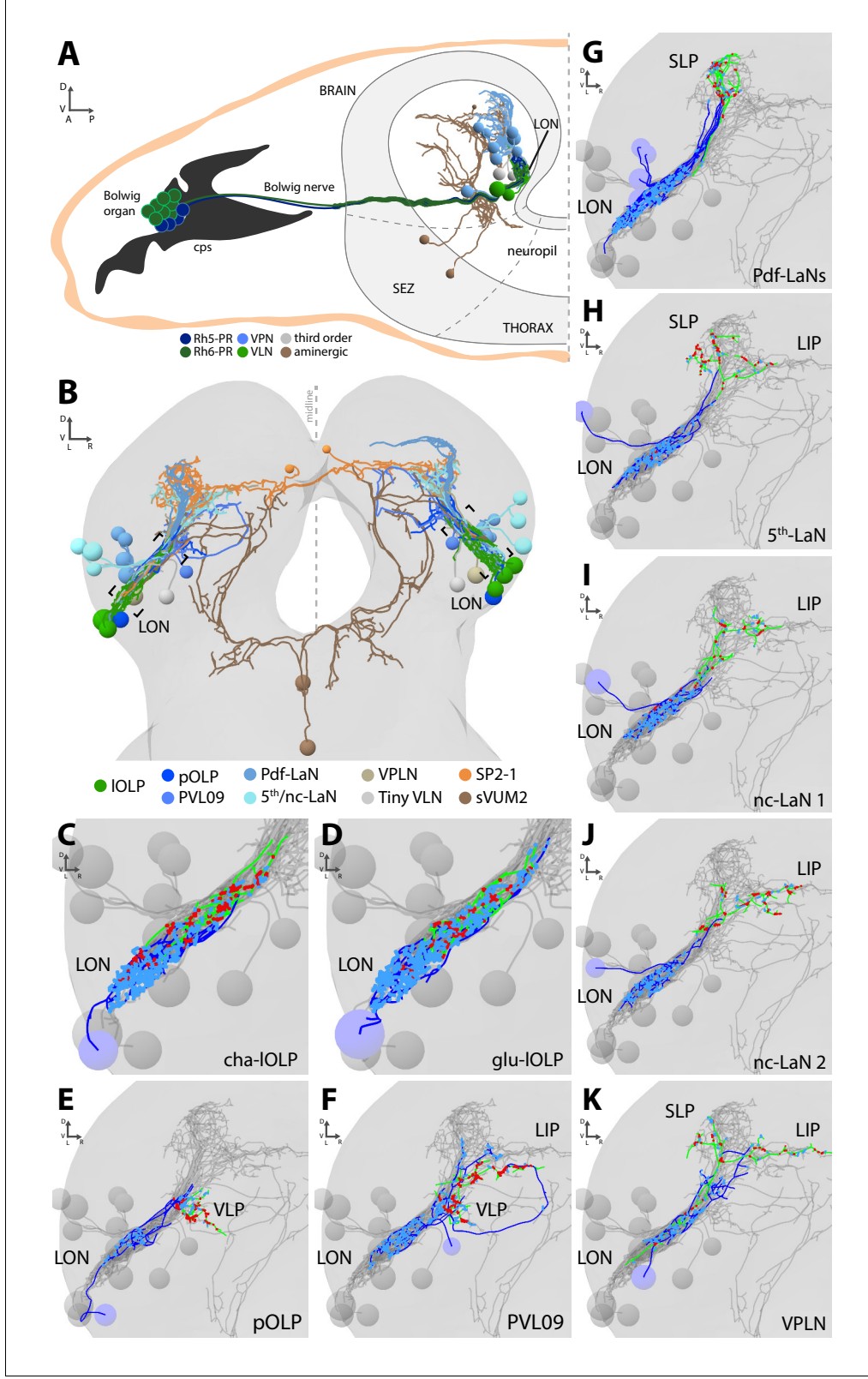

**Figure 1.** Overview of the larval optic neuropil. (**A**) Schematic of the larval visual system with EM-reconstructed skeletons of all larval optic neuropil (LON) neurons. The Rh5-PRs (dark blue) and Rh6-PRs (dark green) cell bodies form the Bolwig organ sitting in the cephalopharyngeal skeleton (cps). They extend their axons to the brain via the Bolwig nerve. In the brain, neurons cell bodies are in the outer layer (gray) and project neurits into the neuropil. *Figure 1 continued on next page*

*Figure 1 continued*

We can distinguish four main classes of neurons: visual projection interneurons (VPN, blue), visual local interneurons (VLN, green), third-order interneurons (gray) and aminergic modulatory neurons (brown). Octopaminergic/tyraminergic modulatory neurons cell bodies sit in the subesophageal zone (SEZ). (**B**) 3D reconstruction of all LON-associated neurons from the ssTEM dataset in both hemispheres (except Bolwig nerves): VLN in green: local optic lobe pioneer neurons (lOLPs); VPN in shades of blue: the projection OLP (pOLP), a novel neuron which is located in the posterior ventral lateral cortex (PVL09), the Pdf-lateral neurons (Pdf-LaNs), the 5th-LaN and the non-clock-LaNs (nc-LaNs); third-order neurons: VPLN in light brown and Tiny VLN in gray; aminergic modulatory neurons: serotonergic neuron (SP2-1, orange) and SEZ-ventral-unpaired-medial-2 octopaminergic/tyraminergic neurons (sVUM2, brown). Posterior view. (**C–K**) 3D reconstructions from ssTEM dataset, posterior view, dendrites in blue, axons in green, presynaptic sites in red, postsynaptic sites in cyan, other LON neurons in gray. VLP: ventral lateral protocerebrum. SLP: superior lateral protocerebrum. LIP: lateral inferior protocerebrum. Both lOLPs (cha-lOLP (**C**) and glu-lOLP (**D**)) have dense arborizations within the LON. (**E**) The pOLP cell body is situated with the lOLP and projects to the VLP. (**F**) PVL09 cell body is situated postero-ventro-laterally to the LON and has an axon with a characteristic loop shape, extending first towards the ventro-medial protocerebrum, then towards the LIP before curving down back to the VLP, where it forms most of its synaptic output. (**G**) The four Pdf-LaNs project to the SLP. (**H**) The 5th-LaN projects both to the SLP and the LIP region, whereas nc-LaN 1 and 2 (**I** and **J**) mainly project to the LIP. (**K**) The third-order neuron VPLN cell body is situated anteriorly to the LON and it has an axon coming back in the LON in top of its projections within both SLP and LIP regions.

DOI: https://doi.org/10.7554/eLife.28387.002

The following video and figure supplements are available for figure 1:

**Figure supplement 1.** Neurotransmitter expression in the LON.
DOI: https://doi.org/10.7554/eLife.28387.003

**Figure supplement 2.** Connections and anatomy of the small third-order neuron Tiny VLN.
DOI: https://doi.org/10.7554/eLife.28387.004

**Figure supplement 3.** : 3D reconstructions of each VPN relative to the lateral horn and to the mushroom body.
DOI: https://doi.org/10.7554/eLife.28387.005

**Figure 1—video 1.** Animation movie of the 3D reconstruction of the larval visual system with all 60 LON neurons. Color code as in *Figure 1B*.
DOI: https://doi.org/10.7554/eLife.28387.006

**Figure 1—video 2.** 3D animations of the rotating brain with all interneurons of the LON in the left hemisphere, first together and then individually, relative to the lateral horn shown by displaying olfactory projection interneurons in red and to the mushroom body as a blue mesh.
DOI: https://doi.org/10.7554/eLife.28387.007

VPNs include the third neuron belonging to the OLPs (*Tix et al., 1989*), which we named accordingly projection-OLP (pOLP, *Figure 1E*), the previously identified pigment dispersing factor (Pdf)-expressing lateral neurons (Pdf-LaNs, *Figure 1G*) and the Pdf-negative 5th-LaN of the circadian clock circuit (*Figure 1H*, *Kaneko et al., 1997*). In addition, in the VPN group we newly identified two non-clock lateral neurons (nc-LaNs) originating from the same neuroblast lineage as the 5th-LaN (*Figure 1I,J*; *Figure 1—figure supplement 1*), and one neuron defined by its 'postero-ventro-lateral' cell body position, termed PVL09 (*Figure 1F*). All these VPNs, except the four peptidergic Pdf-LaNs, are cholinergic (*Figure 1—figure supplement 1*).

We also identified two third-order interneurons that make connections within the LON but do not receive direct inputs from PRs. The first one, is defined by prominent axonal projections beyond the LON and significant pre-synaptic termini within the LON. We therefore named this neuron visual projection-local interneuron (VPLN, *Figure 1K*). We found that the VPLN is glutamatergic (*Figure 1—figure supplement 1*). The second third-order interneuron is a VLN that we named Tiny VLN because of its small size in the current dataset (*Figure 1—figure supplement 2*; no additional information could be collected as no known GAL4 line labels this cell).

Finally, LON circuits are modulated from the central brain by a bilateral pair of serotonergic neurons and two ventral-unpaired-medial neurons of the subesophageal zone that are octopaminergic/tyraminergic and that project bilaterally to both LONs (see below, *Huser et al., 2012*; *Rodriguez Moncalvo and Campos, 2009*; *Selcho et al., 2014*). These neurons match in number and neuromodulator type with the left-right pair of serotonergic neurons and the two bilateral octopaminergic neurons of the larval antennal lobe (*Berck et al., 2016*), providing support for an ancestral

common organization of the visual and the olfactory sensory neuropils (*Strausfeld, 1989*; *Strausfeld et al., 2007*).

With the exception of the unpaired octopaminergic/tyraminergic neurons, we identified in all cases pairs of bilaterally homologous VLNs and VPNs. In addition, in the right brain hemisphere we found an additional fourth OLP, which, together with the variable number of PRs, suggests that the circuit architecture can accommodate a variable number of neurons (see below).

## The larval optic neuropil is organized in three layers

Visual circuits in the mammalian retina as well as in the optic ganglia of the adult fruit fly are organized in layers. These layers are characterized by dendritic arborizations or axonal termini of specific neuron types (*Sanes and Zipursky, 2010*). In *Drosophila* larvae, the LON can be subdivided into three distinct layers based on the innervation of the PR subtypes, which allows to distinguish them in the ssTEM dataset (*Sprecher et al., 2011*). Briefly, Rhodopsin6-PRs (Rh6-PRs) terminate in the distal, most outer layer of the LON (LONd), whereas Rhodopsin5-PRs (Rh5-PRs) terminate in the intermediate LON layer (LONi). The most proximal, inner layer of the LON (LONp) lacks direct PRs input (*Figure 2A–C*).

The layered arrangement of PRs' axon terminals translates into specific connectivity with LON neurons. Most VPNs, whose dendrites do not reach the LONd, receive direct inputs from Rh5-PRs only, whereas the dendrites of Pdf-LaNs span both the LONd and LONi, integrating inputs from both Rh5- and Rh6-PRs (see below; *Figure 1E–K*). The absence of PRs' axon terminals in the LONp deprives the third-order interneurons, VPLN and Tiny VLN, whose dendrites are restricted to the LONp, from any direct PRs inputs (*Figures 1K* and *2F*). Intriguingly, the Tiny VLN integrates inputs within the LONp and projects back to both LONi and LONd (*Figure 1—figure supplement 2*).

Beyond the LON, VPNs' axons target three distinct protocerebral areas, namely the superior lateral protocerebrum, the lateral inferior protocerebrum and the ventro lateral protocerebrum (*Figure 2C,D*). Interestingly, these areas overlap in parts with the lateral horn, involved in innate behaviors, and the mushroom body calyx, involved in associative memory (see below). Within the LON, the dendrites of VPNs are mainly postsynaptic, whereas their axons, upon reaching higher brain areas, present both presynaptic and postsynaptic sites (*Figure 1E–K*). This suggests that VPNs outputs are modulated by input from non-visual neurons, similarly to how olfactory projection interneurons receive non-olfactory inputs (*Berck et al., 2016*). In particular pOLP and the VPLN receive up to 30% of their inputs from non-LON neurons (*Figure 2E*).

## Two light input pathways: each PR subtype targets distinct VPNs and VLNs

Previous studies suggested that only Rh5-PRs are critical for rapid light avoidance, while Rh6-PRs appeared non-essential (*Keene et al., 2011*; *Kane et al., 2013*). However, for entrainment of the molecular clock either PR-type by itself is sufficient (*Keene et al., 2011*). These findings lead us to speculate that Rh5-PRs and Rh6-PRs connect to distinct types of visual interneurons. Supporting this notion, we found that Rh6-PRs synapse principally onto VLNs (79%) and much less onto VPNs (15%, all onto one single VPN type: the Pdf-LaNs; *Figure 2E*, *Figure 2—source data 1*). Conversely, Rh5-PRs preferentially synapse onto VPNs (90%) and much less onto VLNs (6%). An exception to this segregation of PRs inputs are the Pdf-expressing LaNs of the clock circuits. Indeed, the four Pdf-LaNs are the only interneurons that receive direct inputs from both Rh6-PRs and Rh5-PRs (20% from Rh6-PRs and 15% from Rh5-PRs, *Figures 2E* and *3B*), supporting behavioral evidence that either PR-subtype may entrain the larval clock (*Keene et al., 2011*).

Rh6-PRs target the two main VLNs of the LON: the cha- and glu-lOLPs (*Figures 2E,F* and *3A*). Furthermore, the lOLPs main inputs come from Rh6-PRs: up to 75% of cha-OLP input and 58% of glu-lOLP input. Importantly, these two VLNs synapse onto most VPNs including the four Pdf-LaNs and the 5th-LaN of the circadian clock, the nc-LaNs, PVL09 and the third-order interneuron VPLN (*Figure 3C*). pOLP is the only VPN that does not receive inputs from its sister cells, the two lOLPs, and therefore create a direct output pathway of the Rh5-PRs light dependent information towards higher brain regions (*Figure 3B,C*). pOLP is strongly interconnected through axo-axonic connections with PVL09 suggesting that they may reciprocally cross-modulate their synaptic output (*Figures 2E* and *3C*; *Figure 2—source data 2*). Also, the 5th-LaN and the two nc-LaNs, present different

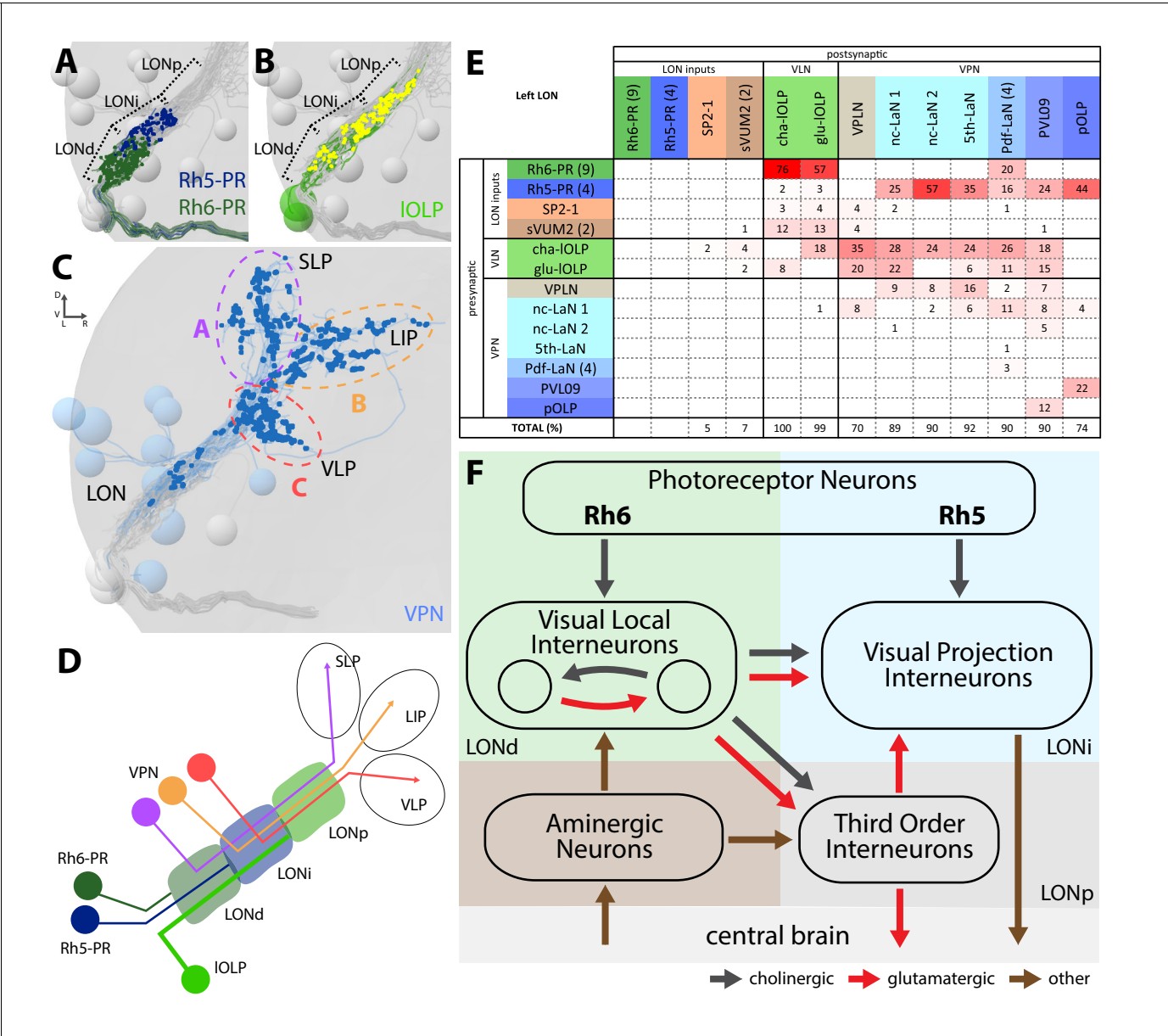

**Figure 2.** The structural organization of the larval optic neuropil. (**A-D**) 3D representations of the presynaptic sites of LON neurons in the left lobe, posterior view. (**A**) Rh6-PRs presynaptic terminals (dark green) define a distal LON layer (LONd) while Rh5-PRs presynaptic connections (dark blue) define an intermediate LON layer (LONi). A third layer of the LON, more proximal (LONp) is devoid of PR terminals. Other LON neurons in gray. (**B**) All LON layers, including the LONp, contain presynaptic sites from the lOLPs (skeletons in green, synapses in yellow). VPNs, Tiny VLN and Bolwig nerve in gray. (**C**) VPNs (blue) make synaptic connections in three main regions outside the LON. VPNs projections define three domains: dorsal domain (A, violet) defined by projections in the superior lateral protocerebrum (SLP), lateral domain (B, orange) in the lateral inferior protocerebrum (LIP), ventral domain (C, red) in the ventral lateral protocerebrum (VLP). VLNs and Bolwig nerve in gray. (**D**) Schematic of the LON three layers: LONd innervated by Rh6-PRs (dark green), LONi innervated by Rh5-PRs (dark blue) and LONp innervated by lOLPs (green); and of the three domains outside the LON were different VPNs subtypes project to (violet, orange and red empty circles). lOLPs also make presynaptic connections in the LONd and LONi (thick line). (**E**) Connectivity table of the left LON with the percentage of postsynaptic sites of a neuron in a column from a neuron in a row. Neurons of same type are grouped, in brackets number of neurons in the group. Same colors as in *Figure 1B*. Only connections with at least two synapses found in both hemispheres were used. (**F**) Simplified diagram of the larval visual system. PRs inputs are cholinergic and define two pathways. Rh5-PRs target VPNs (blue area, LONi) while Rh6-PRs target the two main larval VLNs (green area, LONd). Between these two VLNs, one is cholinergic while the other one is glutamatergic and they both inputs onto VPNs. These VLNs also integrate aminergic modulatory inputs (brown area) that potentially bring information from the central brain. In the LONp, the two third-order neurons receive from the lOLPs. The third-order neuron VPLN, which is glutamatergic, additionally receives from the serotonergic neuron, connects onto the VPNs and also projects towards the central brain (gray area) like other VPNs. Black arrows: cholinergic connections. Red arrows: glutamatergic connections. Brown arrows: connections with other neurotransmitters.

*Figure 2 continued on next page*

*Figure 2 continued*

DOI: https://doi.org/10.7554/eLife.28387.008

The following source data is available for figure 2:

**Source data 1.** Complete synaptic connection matrices from both LONs.

DOI: https://doi.org/10.7554/eLife.28387.009

**Source data 2.** Main connection types for both LONs.

DOI: https://doi.org/10.7554/eLife.28387.010

fractions of inputs from the glu-lOLP, suggesting that they will encounter different levels of

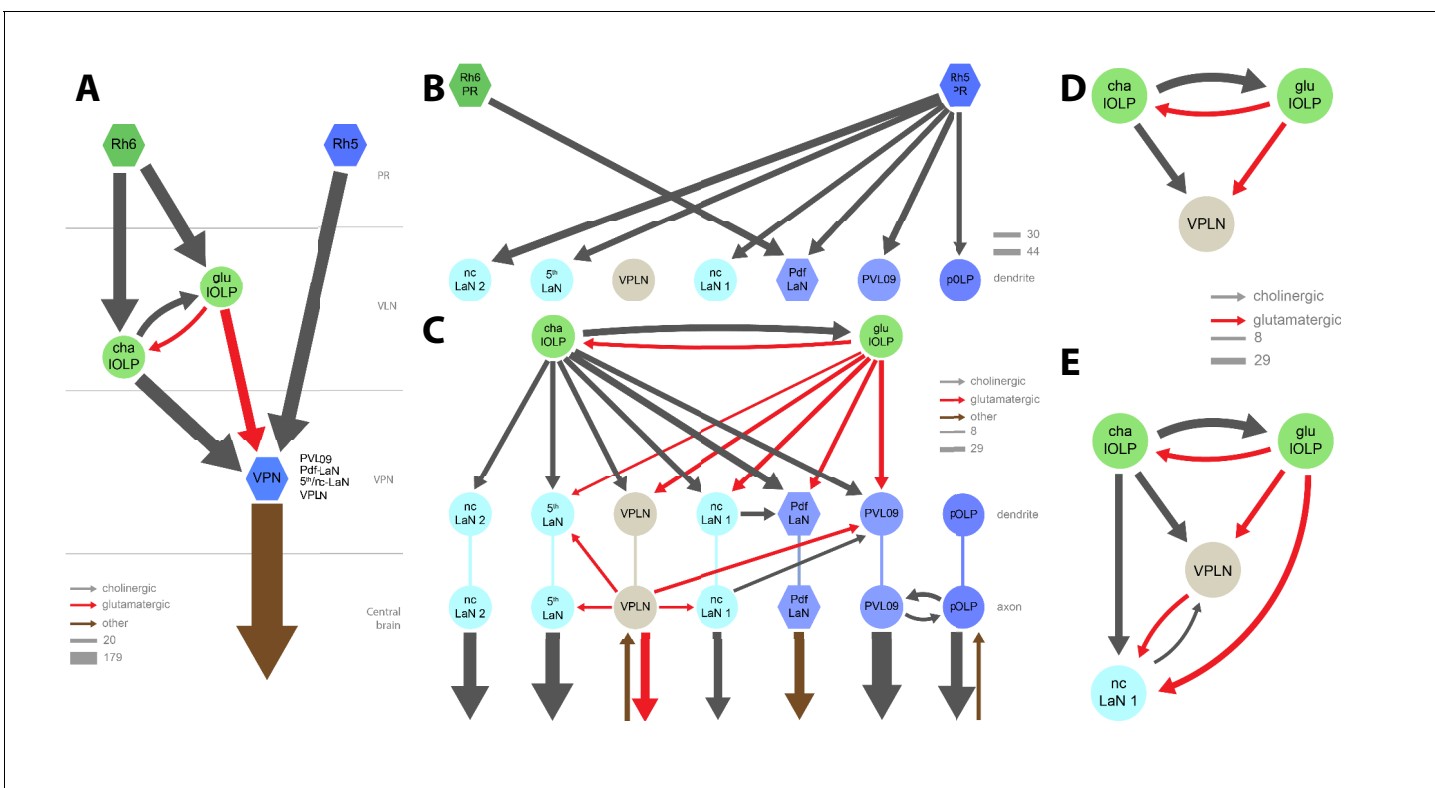

**Figure 3.** Connectivity and circuit motifs of the larval visual system. Left hemisphere, hexagons represent group of cells, circles represent single cell, arrow thickness weighted by the square root of the number of synapses, arrow thickness scale shows minimum and median. Black arrows: cholinergic connections. Red arrows: glutamatergic connections. Brown arrows: connections with other neurotransmitters. VPNs in shades of blue for Rh5-PRs targets plus the VPLN in light brown. (**A**) Wiring diagram of both lOLPs (light green circles). The two lOLPs receives from Rh6-PRs (dark green) and are reciprocally connected. They also share the same type of targets: VPNs (blue) including PVL09, all LaNs and the VPLN, that are direct targets of Rh5-PRs (dark blue) (except the VPLN) and are outputs of the LON towards the central brain. (**B**) The four Pdf-LaNs are the only VPNs that receive from both Rh6-PRs and Rh5-PRs. The VPLN is a third-order neuron that does not receive any inputs from PRs. All other VPNs receive visual inputs uniquely from Rh5-PRs. All inputs from PRs onto VPNs are situated on the target dendrites. (**C**) VPNs, except pOLP, are targets of the two lOLPs and these connections are situated on the VPNs dendrites. Additionally, PVL09 receives inputs from both the VPLN and nc-LaN one while the Pdf-LaNs receive only from the nc-LaN 1, and the 5th-LaN receive only from the VPLN. PVL09 and pOLP are reciprocally connected on their axons. All VPNs transfer light information to neurons deeper in the brain. The VPLN and pOLP additionally receive on their axons some inputs from other neuronal circuits. (**D**) Circuit motif of the VPLN receiving from both lOLPs. E: Circuit motif of the nc-LaN one that is under regulations from cha-lOLP, glu-lOLP and the VPLN. Moreover, nc-LaN one connects back to the VPLN. Similar motifs can be described for other VPNs (*Figure 3—figure supplement 1*).

DOI: https://doi.org/10.7554/eLife.28387.011

The following figure supplements are available for figure 3:

**Figure supplement 1.** Network motifs of the nc-LaN 2, the 5th-LaN, PVL09 and the Pdf-LaNs.

DOI: https://doi.org/10.7554/eLife.28387.012

**Figure supplement 2.** Model of the complete larval visual neural network.

DOI: https://doi.org/10.7554/eLife.28387.013

modulation from this cell (see discussion, *Figures 2E* and *3C*).

The third-order interneuron VPLN, which does not receive direct inputs from PRs (*Figure 3B*), is downstream of the two lOLPs and is itself connecting onto several VPNs including the 5th-LaN, the two nc-LaNs and PVL09 creating another layer of possible computation (see discussion, *Figure 3C–E*).

In summary, most VPNs that directly integrate Rh5-PRs light dependent information may be modulated indirectly by the Rh6-PRs light dependent information via the two VLNs, cha- and glu-lOLPs (see discussion, *Figures 2F* and *3A*).

## VPNs target different brain areas

Distinct areas of the protocerebrum are innervated by the six unique VPNs of the larval visual circuit (the pOLP, the 5th-LaN, the two nc-LaNs, the third-order interneuron VPLN, and PVL09) and by the four Pdf-LaNs of the circadian clock.

pOLP targets the lower lateral horn, an area also innervated by the multiglomerular olfactory projection interneuron (mPN) Seahorse (*Berck et al., 2016*; *Figure 1E*, *Figure 1—figure supplement 3*) with whom it shares numerous postsynaptic partners (data not shown). Since mPN Seahorse integrates inputs from the aversive OR82a-expressing olfactory receptor neuron (*Kreher et al., 2008*), downstream neurons of pOLP and mPN Seahorse are likely contributing to aversive behavior.

Three sister VPNs (the 5th-LaN and both nc-LaN 1 and 2) present similar axon trajectories, dropping synapses in the lateral horn until reaching the accessory calyx of the mushroom body (*Figure 1H–J*, *Figure 1—figure supplement 3*), where they synapse onto Kenyon cells (*Eichler et al., 2017*). On top of its local connections and potential local function (see discussion), the VPLN also has projections beyond the LON in a similar pattern as both nc-LaNs (*Figure 1K*, *Figure 1—figure supplement 3*).

PVL09 is unique among the VPNs in presenting a bifurcated axon with one branch following the other VPNs into the lateral horn and the other branch taking a long looping path below the mushroom body before coming back to the same region as pOLP (*Figure 1F*, *Figure 1—figure supplement 3*). Like most VPNs, PVL09 is under the control of the two lOLPs and the VPLN, but additionally receives inputs from both nc-LaNs on its dendrites and axon (*Figure 3—figure supplement 1*; *Figure 2—source data 2*), suggesting that it integrates broadly all light information.

In summary, all six unique VPNs form synapses in the lateral horn, and three of them (the 5th-LaN, nc-LaN one and nc-LaN 2) synapse onto both the lateral horn and the mushroom body Kenyon cells (*Eichler et al., 2017*). VPN connections onto Kenyon cells may underlie the larval ability to form associative memories with light as a conditioned stimulus, whereas light as an unconditioned stimulus could be encoded via their connections onto the lateral horn (*von Essen et al., 2011*).

Finally, the Pdf-LaNs, necessary for circadian rhythm, project to a region dorsal and more medial to the lateral horn, similarly as in adult *Drosophila* (*Yasuyama and Meinertzhagen, 2010*), where they make few small dyadic synapses from boutons rich in dense-core vesicles (*Figure 1G*, *Figure 1—figure supplements 1* and *3*). Pdf-LaNs boutons also contain clear vesicles suggesting that they might co-express a neurotransmitter, which in adult flies has been suggested to be glycine (*Figure 1—figure supplement 1*; *Frenkel et al., 2017*).

## Central brain feedback via octopaminergic/tyraminergic and serotonergic neurons

Similarly to other sensory modalities (*Roy et al., 2007*; *Dacks et al., 2009*; *Huser et al., 2012*; *Selcho et al., 2014*; *Majeed et al., 2016*; *Berck et al., 2016*), a set of aminergic neurons provide feedback from the central brain into the LON, thus creating an entry point to modulate visual information processing. Both types of modulatory neurons have previously been identified in the LON (*Rodriguez Moncalvo and Campos, 2005*; *Huser et al., 2012*; *Selcho et al., 2014*).

A pair of serotonergic neurons belonging to the SP2 cluster (named SP2-1) connects to the contralateral LON, while receiving presynaptic input predominantly in the ipsilateral protocerebrum (*Figure 4A*, *Figure 4—figure supplement 1*). The two other aminergic input neurons are the octopaminergic/tyraminergic subesophageal zone-ventral-unpaired-medial 2 neurons of the maxillary and mandibular clusters (sVUM2mx and sVUM2md, *Figure 4B,C*, *Figure 4—figure supplement 1*). Each sVUM2 innervates both brain hemispheres in a symmetric fashion.

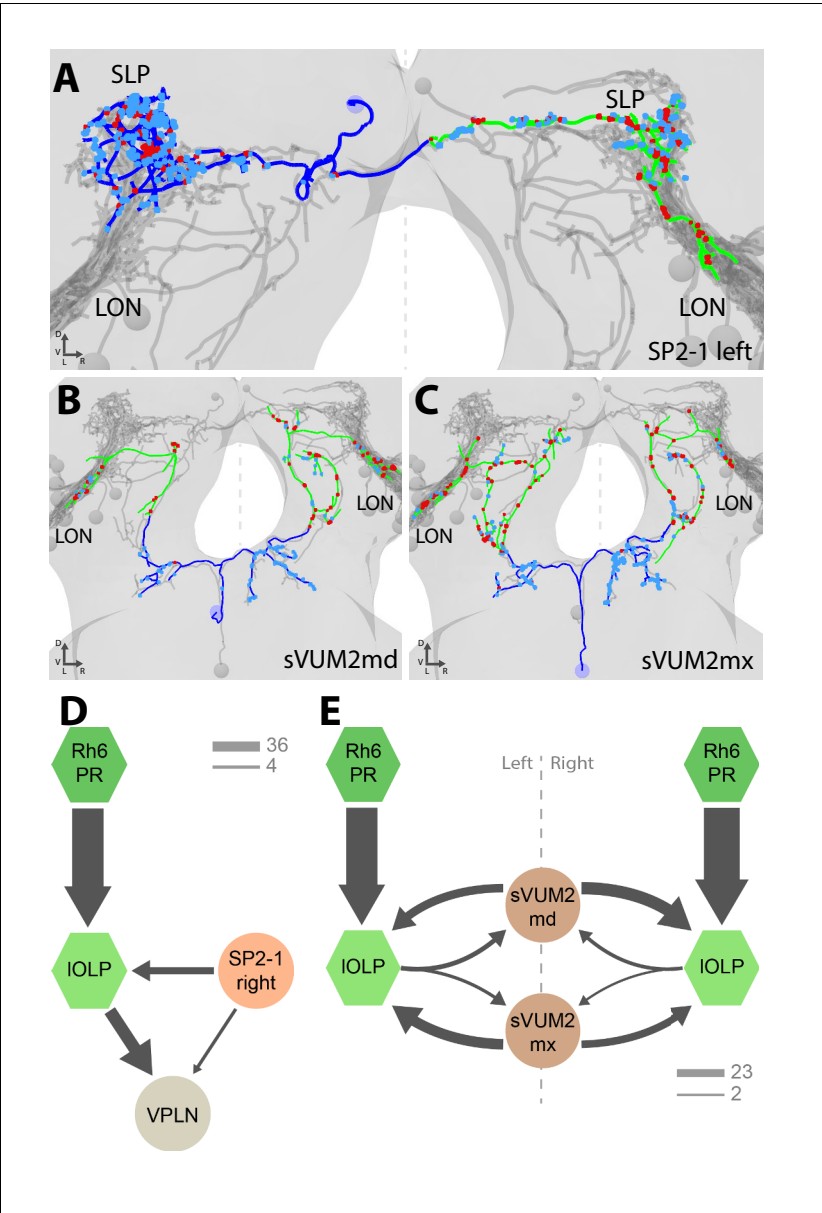

**Figure 4.** Aminergic modulatory inputs of the larval visual system. (A-C) 3D reconstructions from ssTEM dataset, posterior view, dendrites in blue, axons in green, presynaptic sites in red, postsynaptic sites in cyan, other LON neurons in gray, dashed line represent brain midline. (A) The SP2-1 neuron from the left hemisphere innervates the ipsilateral SLP and the contralateral LON. sVUM2md (B) and sVUM2mx (C) neurons are located along the midline in the SEZ with their neurit splitting and innervating both hemispheres in a symmetric fashion. Their bilaterally symmetrical branches receive synaptic input in the SEZ and extend their axon towards the protocerebrum prior to turning laterally and entering the LON. Branches within the protocerebrum and LON contain presynaptic and postsynaptic sites. (D) Connectivity graph showing the SP2-1 neuron (orange) of the right hemisphere connecting with the lOLPs (light green) and the VPLN (light brown) of the left hemisphere. Connections between the lOLPs and the VPLN are also displayed, as well as lOLPs inputs from Rh6-PRs (dark green). (E) Connectivity graph of sVUM2mx and sVUM2md (brown) showing that their only partners are the lOLPs (light green) but in both hemispheres. (D–E) Hexagons represent group of cells, circles represent single cell, arrow thickness weighted by the square root of the number of synapses, arrow thickness scales shows minimum and median.

DOI: https://doi.org/10.7554/eLife.28387.014

The following figure supplement is available for figure 4:

**Figure supplement 1.** Confocal microscopy of the three aminergic neurons of the LON.

DOI: https://doi.org/10.7554/eLife.28387.015

Each SP2-1 neuron connects to the main VLNs (cha- and glu-lOLPs) as well as to the third-order interneuron VPLN of the contralateral side (*Figure 4D*). The octopaminergic/tyraminergic sVUM2 neurons uniquely synapse onto the two lOLPs but in both hemispheres simultaneously (*Figure 4E*). In contrast to the SP2-1, the lOLPs form feedback synapses onto the axonal termini of both sVUM2 neurons. This feedback motif may allow local tuning of the octopaminergic/tyraminergic modulatory input, whereas the serotonergic input is not altered within the LON.

In summary, SP2-1 and sVUM2 mediate feedback from other brain areas to potentially modulate the activity of the two lOLPs of the Rh6-PRs-VLNs pathway and, in case of the SP2-1, may further affect the VPLN. These possible modulations arise from monosynaptic connections between the aminergic cells with these three visual interneurons, while additional effects might be elicited by volume release of serotonin and octopamine (*Dacks et al., 2009*; *Linster and Smith, 1997*; *Selcho et al., 2012*). Further reconstruction is needed to identify the presynaptic inputs of these aminergic modulatory neurons.

## Bilaterally symmetric LON circuits with asymmetric numbers of neurons

Similarly to the non-stereotypic number of ommatidia in the compound eye of the adult fly (*Ready et al., 1976*), the precise number of PRs in each larval eye also varies (*Sprecher et al., 2007*). In the current specimen, we identified thirteen PRs in the left hemisphere and sixteen in the right hemisphere. We found four Rh5-PRs and nine Rh6-PRs in the left hemisphere, and six Rh5-PRs and ten Rh6-PRs in the right hemisphere. Despite this difference in PRs number, homologous LON interneurons in each hemisphere receive a similar fraction of inputs from PRs (*Figure 5—figure supplement 1*), similarly to olfactory projection interneurons that receive an equivalent fraction of inputs from olfactory receptor neuron despite differences in the numbers of these sensory neurons (*Tobin et al., 2017*). This further supports the idea that projection interneurons may regulate the amount of inputs from sensory neurons that they receive relative to the total inputs on their dendrites.

Interestingly, we found a third local-OLP in the right brain hemisphere. Similarly to a non-stereotypic PRs number, variability in OLPs number has been observed before (*Tix et al., 1989*). When using a GAL4 driver labeling glutamatergic neurons (OK371; *Mahr and Aberle, 2005*) we found an extra-glutamatergic-OLP (*Figure 5—figure supplement 1*) at a similar frequency as the presence of the fourth OLP cell had been reported (in about 5% of brains) and displaying an asymmetry between hemispheres. It is unusual to have variability and asymmetries in *Drosophila* neural circuits (*Ohyama et al., 2015*; *Berck et al., 2016*; *Schlegel et al., 2016*; *Jovanic et al., 2016*; *Schneider-Mizell et al., 2016*) but it has been observed before (*Takemura et al., 2015*; *Tobin et al., 2017*; *Eichler et al., 2017*). The presence of an extra-glu-lOLP and variable numbers of PR raises the question of the overall stereotypy of circuit architecture when comparing the left and right LONs.

We analyzed the structure of the left and right LONs circuits with spectral graph analysis of the connectivity matrices by plotting the graph partition metric as a function of the signal flow metric (*Varshney et al., 2011*). Despite the left and right LONs not sharing any interneurons and having a different number of PR and lOLP cells, we observed that neurons of the same type cluster closely together (*Figure 5A*), indicating that the circuit structure in which each identified cell is embedded is very similar in the both hemispheres. The position of the extra-lOLP in this spectral graph analysis plot and its choice of pre- and postsynaptic partners (*Figure 5B*), in particular the many inputs from Rh6-PRs and cha-lOLP, suggest that the extra-lOLP may act as an extra-glu-lOLP, in agreement with its inclusion in the OK371-GAL4 expression pattern. Its reciprocal connections with the Tiny VLN are also in favor of this hypothesis (*Figure 1—figure supplement 2*). Note though that this extra-glu-lOLP also receives some inputs from Rh5-PRs and lacks inputs from the serotonergic neuron unlike other lOLPs, which indicates that it might still act differently than a glu-lOLP. Why some larvae present this additional cell remains to be determined. Importantly, connections among other LON neurons do not seem affected by the presence of an extra-glu-lOLP (*Figure 5B*; *Figure 5—figure supplement 1*). In conclusion, while the number of PRs and VLNs can vary in the larval visual system, the overall circuit architecture is maintained, and in particular the output channels (VPNs) are identical.

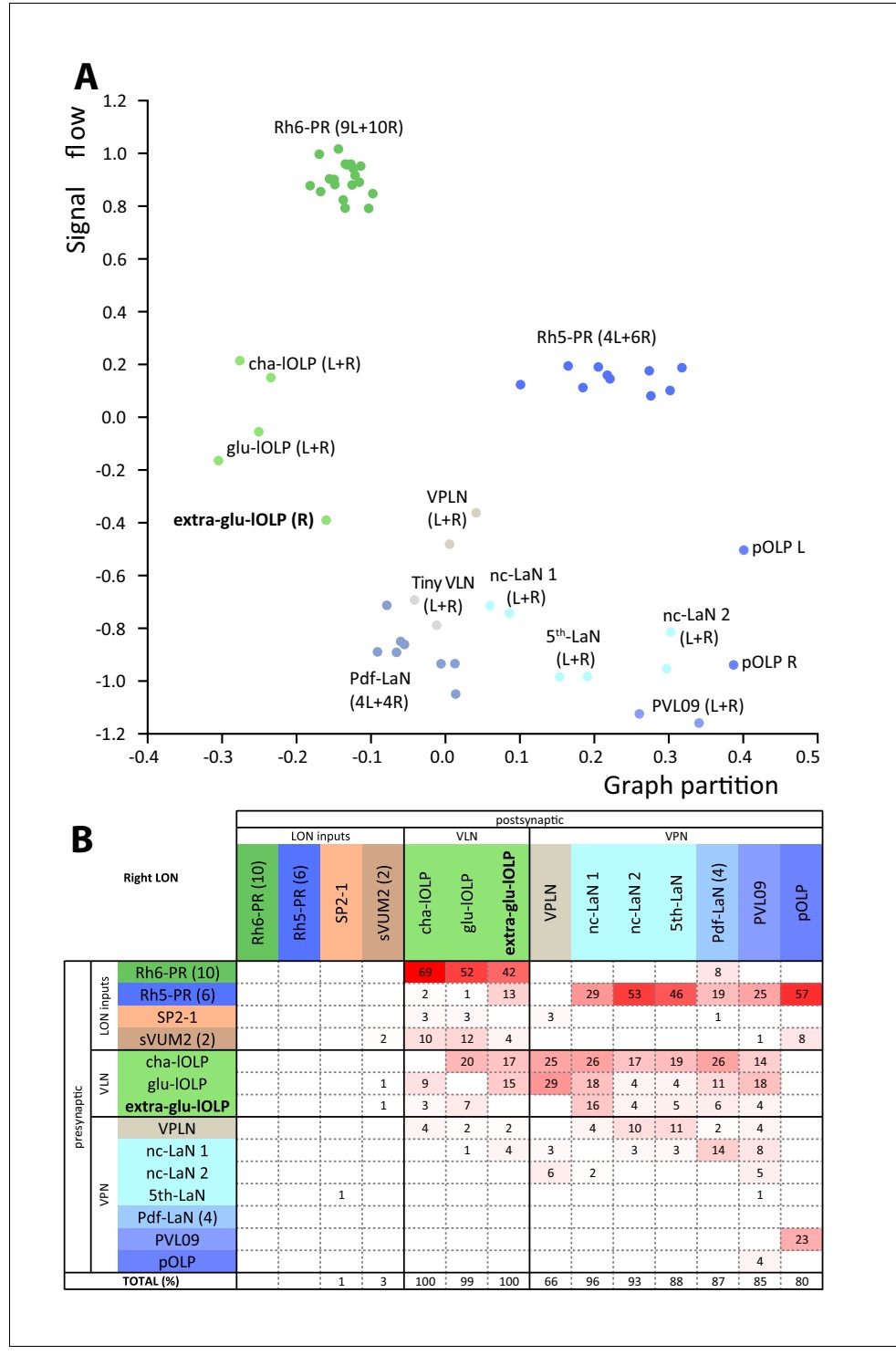

**Figure 5.** Larval optic neuropil architecture is maintained despite a variable number of neurons. Color code as in *Figure 1B*. (**A**) We compared the structure of the left (**L**) and right (**R**) LON circuits with spectral graph analysis. For the left and right connectivity matrices, we plotted the components of the first nonzero eigenvector of the graph Laplacian associated with each neuron (x axis) as a function of the signal flow metric (y axis), measures that reflect the topological role of neurons in the circuit (*Varshney et al., 2011*). We excluded the neuromodulatory neurons SP2-1, which are weakly connected, and the sVUM2md and sVUM2mx, which project bilaterally. Visual information flows from PRs at the top towards VPNs at the bottom. In this representation, bilaterally homologous neurons share a similar position revealing that both sides have similar network topology (*Schneider-Mizell et al., 2016*). Note how the extra-glu-lOLP of the right hemisphere (bold) positioned closely with the other lOLPs. (**B**)

*Figure 5 continued on next page*

*Figure 5 continued*

Connectivity table of the right LON with the percentage of postsynaptic sites of a neuron in a column from a neuron in a row. Only connections with at least two synapses found in both hemispheres were used (except for the extra-glu-lOLP).

DOI: https://doi.org/10.7554/eLife.28387.016

The following figure supplement is available for figure 5:

**Figure supplement 1.** Stability of the connections of LON neurons despite the variability in neuron number.

DOI: https://doi.org/10.7554/eLife.28387.017

## Discussion

A shared characteristic of many visual systems is the retinotopic organization allowing visual processing in a spatially segregated fashion by the transformation of the surrounding environment into a 2D virtual map (for review, *Sanes and Zipursky, 2010*). Within the compound eye of the adult fruit fly, this is achieved by the prominent organization of PRs in ommatidia in the retina that is maintained through underlying cartridges in the lamina and columns in the medulla. The *Drosophila* larval eye lacks ommatidia or a similar spatial organization of PRs. Nevertheless larvae can navigate directional light gradients and form light associative memories using their simple eyes (*Kane et al., 2013*; *von Essen et al., 2011*; *Humberg and Sprecher, 2017*).

In this study, we described the synapse-level connectome of the larval first visual center by reconstructing neurons recursively from the optic nerves to third-order neurons following all chemical synapses in a nanometer-resolution EM volume of the whole central nervous system. We found that the two PR subtypes synapse onto distinct target interneurons, showing a clear separation of visual information flow at the first synapse level. Rh5-PRs predominantly synapse onto VPNs, which transfer light information to distinct regions including the lateral horn and the mushroom body calyx, whereas Rh6-PRs strongly synapse onto two VLNs (cha- and glu-lOLPs). Moreover, the flow of information is convergent as these two main VLNs in turn synapse onto most VPNs. These two main VLNs also receive input from both the serotonergic as well as the octopaminergic/tyraminergic systems, suggesting that their activity may be modulated by input from central brain circuitry. Thus, a key feature of the larval visual circuit is that the Rh6-PRs-pathway feeds into the Rh5-PRs-pathway, suggesting a tuning function for Rh6-PRs and the two VLNs that also integrate external modulatory inputs (*Figure 2F*; *Figure 3—figure supplement 2*). This is not excluding the possibility of electrical connections mediated by gap junctions that are not visible in the ssTEM volume.

### Hypothesized functional interpretation of circuit architecture

#### The Rh5-PRs-VPNs pathway may provide information about the ambient light intensity

Larval PRs responses to light remain currently unknown. Moreover, compared to adult flies PRs, larval PRs lack a clear rhabdomeric structure, do not present intracellular pigment granules and the Bolwig organ is not equipped with focusing optics or other accessory cells (*Green et al., 1993*; *Sprecher et al., 2007*). Nevertheless, larval PRs express the main actors of the phototransduction pathway like adult PRs (*Rosenbaum et al., 2011*; *Mishra et al., 2016*; *Bernardo-Garcia et al., 2016*). It therefore remains difficult to predict how larval PRs response to light. Larval PRs might either adapt and detect the mean light intensity, like adult PRs (for review *Clark and Demb, 2016*), or might detect the absolute light intensity. In either case, one can propose that larval PRs provide information about the ambient light intensity of the larval environment.

In other sensory systems such as the olfactory system of the adult (*Rybak et al., 2016*) and larval *Drosophila* (*Berck et al., 2016*) and in the chordotonal mechanosensory systems of the locust (*Yuan et al., 2011*) and *Drosophila* larva (*Ohyama et al., 2015*; *Jovanic et al., 2016*) axonal terminals of sensory neurons receive abundant inhibitory inputs from central neurons. Such presynaptic inhibition of sensory terminals may be employed to mediate lateral inhibition (*Wilson and Laurent, 2005*; *Olsen and Wilson, 2008*), to implement divisive normalization (*Olsen et al., 2010*) or to encode a predicted future stimulation (*Wolf and Burrows, 1995*). However, in the larval visual circuit, we do not observe significant synaptic connections onto the PRs (*Figures 2E* and

5B), suggesting that the light information encoded by PRs is passed on with no alteration at the level of the first synapse.

We therefore propose that larval Rh5-PRs may provide information about the ambient light intensity directly onto the larval visual system output neurons, the VPNs.

## The Rh6-PRs-VLNs pathway may compute variations in light intensity

Compared to Rh5-PRs that directly connect the VPNs, the Rh6-PRs inputs are relayed to the VPNs only via the two VLNs, the cha- and glu-lOLPs. In dissociated larval Pdf-LaNs, it was shown that applying acetylcholine increased their calcium response whereas applying glutamate reduced their calcium response (*Wegener et al., 2004*; *Hamasaka et al., 2007*), suggesting that acetylcholine is an excitatory neurotransmitter for these cells whereas glutamate would be inhibitory. Acetylcholine can act as an inhibitory neurotransmitter through a metabotropic muscarinic receptor (*Shinomiya et al., 2014*; *Ren et al., 2015*), however, it more often acts as an excitatory neurotransmitter in *Drosophila* (*Baines and Bate, 1998*; *Burrows, 1996*). If both cha and glu-lOLPs were to be excitatory, the positive feedback loop between these two cells would likely result in an enhanced signal propagation of Rh6-PRs inputs towards VPNs. Such a positive feedforward feature downstream of the two PR-subtypes could make sense if they detected qualitatively distinct visual features such as spectral cues or intensity. If for instance Rh6-PRs responded to a vastly distinct light-spectrum than Rh5-PRs or if Rh6-PRs were low-light sensitive, this positive feedforward loop could result in spectral integration or amplification of low-light input. However, absorption spectra of both larval PRs are largely overlapping and there is currently no evidence for distinct detection capacities of both PRs (Salcedo et al., 1999; *Humberg and Sprecher, 2017*). Therefore, in our view a more favorable scenario is that outputs from cha-lOLP are likely excitatory while the outputs from glu-lOLP are likely inhibitory, similarly as glutamate has been found to mediate inhibition in the adult fly lobula plate (*Mauss et al., 2014*) and in the antennal lobe (*Liu and Wilson, 2013*; *Berck et al., 2016*). Therefore, in *Figures 2F* and *3* one could interpret cholinergic arrows (black) as excitatory and glutamatergic arrows (red) as inhibitory (also see *Figure 3—figure supplement 2*).

If we consider cha-lOLP as excitatory and glu-lOLP as inhibitory, then Rh6-PRs signals appears to be positively transferred to VPNs only via the cha-lOLP (*Figure 3A*). In addition, cha-lOLP then receives strong inputs from the inhibitory glu-lOLP (*Figures 2E* and *3A*), which is also almost exclusively driven by Rh6-PRs inputs. Therefore, this inhibition of glu-lOLP onto cha-lOLP could mediate a form of indirect presynaptic inhibition of the Rh6-PRs inputs. This motif made by the cholinergic Rh6-PRs driving both the cha-lOLP and the glu-lOLP, and with the glu-lOLP inhibiting the cha-lOLP, creates then an incoherent feedforward loop (*Alon, 2007*). Therefore this motif could make the cha-lOLP responsive to the derivative of Rh6-PRs activity, that is to the variations in light intensity. As cha-lOLP VLN is likely to be a positive relay of Rh6-PRs inputs onto VPNs, VPNs could therefore respond to increments in light intensity (ON response).

Moreover, cha-lOLP inputs onto the glu-lOLP and both cha- and glu-lOLPs further connect onto the VPNs. Considering cha-lOLP as excitatory and glu-lOLP as inhibitory, this creates a second incoherent feedforward motif making VPNs potentially responsive to the derivative of cha-lOLP inputs. As cha-lOLP inputs may already represent the first derivative of Rh6-PRs inputs, this second incoherent feedforward motif may therefore make VPNs responsive to the acceleration of light intensity raises.

In summary, glu-lOLP may provide both presynaptic inhibition of Rh6-PRs inputs by inhibiting the relay neuron cha-lOLP, and postsynaptic inhibition by directly inhibiting most VPNs. Consequently, VPNs could respond to the ambient light intensity (from Rh5-PRs) and to the variations of light intensity (from the Rh6-PRs-VLNs pathway). Interestingly, olfactory projection interneurons in the adult *Drosophila* also respond to a mixture of odorant concentration and of the acceleration of odorant concentration (*Kim et al., 2015*). Comparable to other sensory system (*Klein et al., 2015*; *Schulze et al., 2015*), a measure of changes in light intensity over time would enable light gradient navigation (*Kane et al., 2013*; *Humberg and Sprecher, 2017*).

## Phasic inhibition could sharpen ON and OFF responses

In the wiring diagram, we found a glutamatergic (therefore putatively inhibitory) third-order interneuron, named VPLN and mainly driven by cha- and glu-OLPs. The timing of cha- and glu-lOLPs

input may be critical for the VPLN function. If cha-lOLP input precedes glu-lOLP input, the VPLN may potentially be activated only upon an increase in light intensity via Rh6-PRs (*Figure 3D*). This is unlike the other glutamatergic interneuron, glu-lOLP, which receives direct inputs from PRs and therefore can be tonically active in the presence of light. The VPLN specifically synapses onto multiple VPNs (the 5th-LaN, the two nc-LaNs and PVL09) and therefore potentially inhibits these cells that also receive strong inputs from glu-lOLP (*Figure 3C,E*, *Figure 3—figure supplement 1*). In consequence, these VPNs may be subject to tonic inhibition (from glu-lOLP) under constant light conditions and to phasic inhibition (from the VPLN) upon an increase in Rh6-PRs dependent light intensity. Therefore, phasic inhibition from the VPLN may potentially refine the temporal resolution of VPNs responses to increment of light intensity (ON response).

An interesting aspect resulting from tonic inhibition of VPNs during constant light stimulation is what happens when the light intensity decreases (OFF response). Some neurons subject to tonic inhibition can have a rebound of activity when inhibition is lifted (*Marder and Bucher, 2001*; *Hedwig, 2016*). If, via this mechanism, VPNs were to increase their activity upon a decrease in light intensity, they would be encoding an OFF response. Potentially all VPNs under tonic inhibition from glu-lOLP could have this rebound of activity after a light intensity decrease. Interestingly, two cholinergic VPNs (the nc-LaN 1 and 2) synapse onto the VPLN that could then become more strongly activated by the OFF response (*Figure 3D*, *Figure 2—source data 1*). As the VPLN is inhibiting several VPNs (the 5th-LaN, the two nc-LaNs and PVL09), this could therefore create a second period of phasic inhibition allowing to maintain these VPNs OFF responses brief.

In conclusion, the connectivity of the third-order glutamatergic VPLN putatively inhibitory suggests that it may refine the temporal resolution of VPNs ON and OFF responses through phasic inhibition.

## The visual circuit supports previous behavioral observations

In addition, this complete wiring diagram of the larval visual system can explain why in previous studies only Rh5-PRs appeared to be required for light avoidance while Rh6-PRs seemed dispensable in most experimental conditions (*Keene et al., 2011*; *Kane et al., 2013*; *Humberg and Sprecher, 2017*). Moreover, Rh6-PRs mutant larvae were shown to avoid light wavelengths from ultra-violet to green, supporting the idea that Rh5-PRs respond to a wider range of wavelength of light than only 'blue' light (*Humberg and Sprecher, 2017*; *Salcedo et al., 2009*). Taken together, our current data and the lack of evidence of color vision capability in larvae, we propose that the two larval PRs are rather specialized in detection of ambient light intensity and of temporal variations of light intensity.

Based on our connectomic analysis, when Rh6-PRs are mutated or functionally silenced, Rh5-PRs may still provide light ambient intensity information, and the cholinergic inputs from nc-LaNs onto the glutamatergic VPLN could inhibit the VPNs, computing on its own some response to changes in light intensity. Blocking the VPLN activity in a Rh6-PRs depletion background would allow to study whether ambient light information is sufficient for visual navigation. Moreover, when Rh5-PRs are disabled, VPNs do not receive ambient light information while potentially being under tonic inhibition from the glu-lOLP, which therefore could shut down their activity. Given that in Rh5-PRs depletion background, larval visual responses are highly defective, this further suggests that ON and OFF responses alone are not sufficient to allow larval light avoidance. We therefore propose that it may need both a baseline activity of VPNs, provided by ambient light intensity information, and the modulations of this baseline activity in response to light intensity variations.

Only the Pdf-LaNs involved in circadian rhythm receive direct inputs from both PRs subtypes and can therefore still receive ambient light information in either PRs' depletion condition. This broad light information integration capacity is in agreement with evidence that both PRs subtypes are sufficient to entrain the larval clock (*Keene et al., 2011*). While further reconstructions of their postsynaptic partners are required, the four Pdf-LaNs appear identical in sensory inputs, local connections and anatomy, raising the question of what such redundancy would allow.

## Comparisons between the larval eye and adult compound eye

The basic organization of the visual system in the larva, the adults fly as well as with the visual circuit of vertebrates show several similarities and possible shared characteristics (*Figure 6*). Several possible interpretations of the photoreceptor types as well as circuit motifs may be derived. One possible

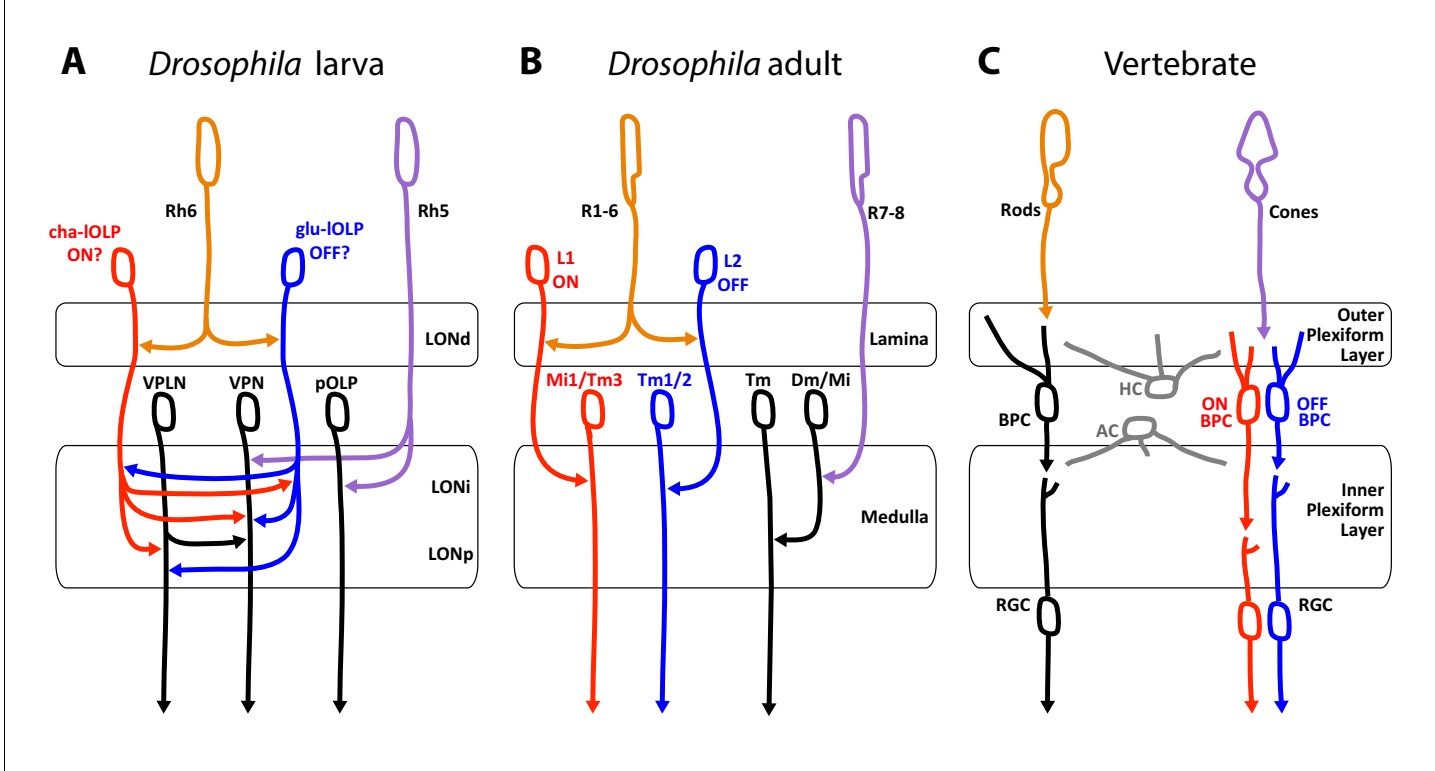

**Figure 6.** Comparison of the structural organization and putative functions of the *Drosophila* larval visual circuit with the *Drosophila* adult compound eye and the vertebrate visual circuit. (**A**) Larval visual circuit as described in this paper. Two main groups of VPNs receive input from Rh5-PRs (purple): one corresponds to the pOLP that only receives inputs from Rh5-PRs, whereas the second group (VPN) also receives inputs from cha and glu -lOLPs (red and blue) that are targets of Rh6-PRs (orange). The third-order neuron VPLN does not receive from either larval PRs but is downstream of the two lOLPs and connect onto the VPN group in top of its projection beyond the LON. We propose that cha and glu -lOLPs control light intensity increment and decrement (ON/OFF) detection respectively and transfer these information to the VPLN and the VPN group. Moreover, the VPLN may potentially keep the ON/OFF responses brief. (**B**) Model of a single unit of the fly compound eye where R1-6 PRs (orange) are well known to be involved in contrast and motion detection, whereas R7-8 PRs (purple) are involved in color sensing (*Sanes and Zipursky, 2010*; *Clark and Demb, 2016*, for reviews). In the lamina, R1-6 PRs make connections to the glutamatergic L1 neuron controlling the ON pathway (red) and to the cholinergic L2 neuron controlling the OFF pathway (blue). In the deeper medulla, L1 and L2 reach their targets (Mi1, Tm1/2), whereas R7-8 PRs connect to medullar neurons (Dm/Mi). (**C**) Model of the vertebrate visual circuit (*Sanes and Zipursky, 2010*; *Clark and Demb, 2016*), for reviews). Cones (purple), which are also the color sensors of the retina, connect to bipolar retinal cells (BPC) which constitute the ON or OFF pathways depending on the glutamate receptor they express (ON BPC and pathway in red, OFF BPC and pathway in blue). Rods (orange) also connect to BPC and control vision in dim light conditions. LONd, LONi and LONp: distal, intermediate and proximal larval optic neuropil. Mi: medulla intrinsic neurons; Tm: transmedulla neurons; Dm: dorsal medulla neurons. RGC: retina ganglion cells; HC: horizontal cells; AC: amacrine cells.

DOI: https://doi.org/10.7554/eLife.28387.018

scenario lies in the comparison between PR types found in the different eye types and their known or putative roles in temporal coding as discussed here.

First, these three visual systems have each two main types of photo-sensory neurons: Rh5- and Rh6-PRs for the *Drosophila* larva, inner and outer PRs for adult flies, cones and rods for vertebrates (*Sprecher et al., 2007*; *Sprecher and Desplan, 2008*; *Friedrich, 2008*; *Sanes and Zipursky, 2010*). Here we also confirmed that the LON was organized in layers (LONd, LONi, LONp) similarly as the adult flies optic lobe (lamina, medulla) and the vertebrate retina (outer and inner plexiform layers) (*Sanes and Zipursky, 2010*, *Figure 6*). Moreover, the mode of development of *Drosophila* larval and adult eyes reinforces similarities between Rh5-PRs with an inner-PR type and Rh6-PRs with an outer-PR type: in the adult, R8-PR precursors are formed first to recruit outer PRs, likewise in the larva, Rh5-PR precursors develop first and then recruit the Rh6-PRs, in both cases via the EGFR pathway (*Sprecher et al., 2007*). Also, development of both inner PRs and Rh5-PRs depend on the transcription factors *senseless* and *spalt* (*Sprecher et al., 2007*; *Sprecher and Desplan, 2008*; *Mishra et al., 2013*). Finally, the inputs from serotonergic and octopaminergic/tyraminergic neurons

onto the LON are also a shared feature with the adult visual system where serotonin has been linked to circadian rhythmicity (*Yuan et al., 2005*) and where visual cues during flight are modulated by octopaminergic inputs (*Suver et al., 2012*; *Wasserman et al., 2015*). In the LON, our data suggest that the serotonergic neuron SP2-1 and octopaminergic/tyraminergic neurons sVUM might modulate the sensitivity to light increment and decrement (ON/OFF) by directly connecting the cha- and glu-lOLPs VLNs. These ON and OFF responses might be further affected by inputs from SP2-1 onto the VPLN.

A potential functional comparison between the larval and the adult visual system emerge from the similar neurotransmitters expressed and the functions proposed for the larval VLNs cha- and glu-lOLPs and the adult glutamatergic L1 and cholinergic L2 interneurons of the lamina (*Figure 6A,B*; *Gao et al., 2008*). In the adult fly, the outer R1-6 PRs connect to L1 and L2 interneurons that convey distinct responses to light increment (ON) and decrement (OFF) (*Joesch et al., 2010*; *Clark et al., 2011*; *Strother et al., 2014*; *Behnia et al., 2014*). The adult PRs are histaminergic and induce hyperpolarization (inhibition) in both L1 and L2 upon light increment, followed by a rebound of activity of these two cells upon light decrement (*Hardie, 1987*; *Reiff et al., 2010*; *Clark et al., 2011*); reviewed in *Borst and Helmstaedter, 2015*). Therefore upon light increment, PRs inhibit the glutamatergic L1, which results in the disinhibition of the downstream targets of L1 (ON response). In turn, upon light decrement PRs inhibit the cholinergic L2 less, which results in the activation of the downstream targets of L2 (OFF response). However, the larval PRs are cholinergic (*Yasuyama et al., 1995*; *Keene et al., 2011*) and, at least for the Pdf-LaNs (*Yuan et al., 2011*), excite their targets activity in response of light. Therefore in larvae, an ON response could result from an increase of excitation from cha-lOLP onto VPNs when light intensity increases (via the increase of Rh6-PRs inputs) and could be kept transient by inhibition from the glu-lOLP (indirect presynaptic inhibition of Rh6-PRs inputs) and by phasic inhibition from the VPLN. In turn, an OFF response could result from the disinhibition of VPNs from glu-lOLP inhibition when light intensity decreases and may also be kept transient by phasic inhibition from the VPLN. Therefore, whereas the glutamatergic L1 conveys the ON response and the cholinergic L2 conveys the OFF response in adult flies, we propose that the glu-lOLP could convey the OFF response and the cha-lOLP could convey the ON response in larvae (*Figure 6A,B*). Another similarity between the two lOLPs and the adult L1 and L2 is that each lOLP receives an equivalent number of inputs from each Rh6-PRs (about 20 synapses in average, *Figure 2—source data 1*), similarly as adult L1 and L2 receives about 50 inputs from each R1-6 PRs (*Meinertzhagen, 1989*), suggesting that the information transfer could have a similarly high fidelity. However, while ON/OFF responses in adult flies and vertebrates (*Figure 6B,C*) are involved in motion detection, such ability within one eye is achieved through downstream direction-sensitive cells that integrate information from several points in space (*Clark and Demb, 2016*). As larval eyes lack ommatidia such capacity seems unlikely, however the ON/OFF detection could already suffice for larval visual navigation (*Kane et al., 2013*; *Klein et al., 2015*; *Schulze et al., 2015*; *Humberg and Sprecher, 2017*). Another important difference between the larval and adult systems is that in the LON the cha- and glu-lOLPs are strongly interconnected and both input onto the same targets, whereas in the adult fly, the L1 and L2 are not chemically connected (only electrically, *Joesch et al., 2010*) and the ON/OFF pathways are separated for two additional synaptic levels before converging on direction-sensitive cells in the lobula plate (*Takemura et al., 2013*; *Clark and Demb, 2016*). Consequently, whereas the adult L1 and L2 have similar responses to light (*Clark et al., 2011*; *Borst and Helmstaedter, 2015*), the reciprocal connections between cha- and glu-lOLPs may uniquely shape the light response of these two cells.

If the Rh6-PRs-VLNs pathway contributes indeed to a function related to light increment or decrement remains hypothetical. Moreover, since Rh5-PRs survive metamorphosis giving rise to the adult Hofbauer-Buchner eyelet, it is possible that the larval visual circuit more closely resembles the eyelet circuitry than the one of the adult compound eye (*Sprecher and Desplan, 2008*).

## Comparison between the visual and the olfactory first-order processing centers

The LON neural network presents many similarities with the larval olfactory wiring diagram (*Berck et al., 2016*), favoring a potential common organizational origin of these sensory neuropils as suggested before (*Strausfeld, 1989*; *Strausfeld et al., 2007*). Odorant cues are perceived by

olfactory receptor neurons that project to the antennal lobe where they contact olfactory projection interneurons and olfactory local interneurons. Similarly to visual circuits, sensory inputs in the antennal lobe are segregated in olfactory receptor specific domains termed glomeruli (*Fishilevich et al., 2005*; *Masuda-Nakagawa et al., 2009*). Most olfactory receptor neurons and olfactory projection interneurons are cholinergic like PRs and VPNs (*Keene et al., 2011*; *Yasuyama and Salvaterra, 1999*; *Python and Stocker, 2002*), and we find glutamatergic, potentially inhibitory, local interneurons in both systems (*Berck et al., 2016*). Reciprocal synapses between cha-lOLP and glu-lOLP in the visual circuit may be functionally equivalent to the connections between some olfactory receptor neurons and the glutamatergic Picky olfactory local interneuron 0 of the antennal lobe, suggesting that this reciprocal motif in the LON could indeed contribute to detecting changes in light intensity by computing the first derivative of the stimulus (*Berck et al., 2016*). Additionally, we can observe presynaptic and postsynaptic inhibition in both sensory systems while the strategies of implementation are somewhat different. Interestingly, we observed that most projection interneurons of both sensory systems may bring a mixture of stimulus intensity and acceleration in stimulus intensity to the lateral horn and to the mushroom body calyx (*Figure 1—figure supplement 3*; *Kim et al., 2015*). Finally, both sensory systems are modulated by aminergic neurons inputs on specifics local interneurons. Interestingly, both the lOLPs VLNs and the olfactory Broad local interneurons Trio form feedback synapses onto the axonal termini of their respective bilateral octopaminergic/tyraminerigc neurons while this is not the case for serotonergic neurons (*Berck et al., 2016*).

### Concluding remarks

Identification of synaptic connectivity and neurotransmitter identity within the larval visual circuit allows us to formulate clear predictions on the response profile and function of individual network components. Based on the circuit map we suggest that the Rh6-PRs-VLNs pathway might be required for the detection of light intensity changes, whereas the Rh5-PRs-VPNs pathway could provide direct ambient light intensity information. In the future, behavioral studies or physiological activity recording of visual circuit neurons will allow to add additional layers onto this functional map.

## Materials and methods

### ssTEM based neuronal reconstruction

In order to reconstruct the larval visual system, we used the serial-section transmission electron microscopy (ssTEM) volume of the entire nervous system of a first instar larva as described in *Ohyama et al. (2015)*. Briefly, a 6-h-old *[iso] CantonS G1 x w1118* brain was serially sectioned at 50 nm and imaged at 4 nm per pixel with an FEI Spirit BioTWIN TEM (Hillsboro). After images processing and compression, the whole dataset was stored on servers accessible by the web page CATMAID (Collaborative annotation Toolkit for Massive Amounts of Image Data, http://openconnecto.me/catmaid/, *Saalfeld et al., 2009*). The reconstruction was performed manually following the method used in *Ohyama et al. (2015)* and described in detail in *Schneider-Mizell et al., 2016*. All photoreceptor neurons were traced from the Bolwig nerve's entrance in the ssTEM stack up to their terminals within the larval optic neuropil. The loss of eight 50 nm serial sections between frames 1103 and 1112 have made difficult to get all the neurons complete especially in the right hemisphere. Therefore, attempts to cross the gap for a neuron were validated with its contralateral homolog. We found 60 neurons in total, measuring 12.5 millimeters of cable and presenting 2090 presynaptic sites and 4414 postsynaptic sites. seven tiny fragments (that amount to 0.018 millimeters of cable and 20 postsynaptic sites in total) could not be attached to full neuronal arbors. The reconstruction required 134 hr plus an additional 43 hr for proofreading.

### Fly strains

Flies were reared on standard cornmeal and molasses food in a 12 hr: 12 hr light-dark cycle at 25°C. We used the following strains for each subset of neurons (number of neurons of interest; neuron's name): from the GMR Rubin GAL4 (R) and JRC split-Gal4 (SS0) collections of the Howard Hughes Medical Institute at Janelia Research Campus: SS01740 (1; serotonergic SP2-1 neuron), SS02149 (2; octopaminergic/tyraminergic sVUM2 neurons), R72A10-GAL4 (3; OLPs), SS01724 (1; glutamatergic lOLP), SS01745 (1; projection OLP), R20C08-GAL4 and SS00671 (1; PVL09), R19C05-GAL4 (3; nc-

LaNs and 5th-LaN; plus the 4 Pdf-LaNs are weakly expressed) and SS01777 (1; VPLN). From Bloomington BDSC: OK371-GAL4 (VGluT-Gal4, ♯26160), and pJFRC29-10xUAS-IVS-myr::GFP-p10 (attP40) (referred as UAS-myr::GFP, ♯32198). Kind gift from B. Egger: w;; UAS-His2B-mRFP/TM3 (*Mayer et al., 2005*).

## Immunohistochemistry

All confocal stacks are from early third instar larvae. All identification of a neuron neurotransmitter expression was performed on first and third instars. Larvae were dissected 4 days after egg laying. Brain dissections where performed in cold phosphate-buffered saline (PBS, BioFroxx, ♯1346LT050, 1X in dH2O). Brains were fixed in 3.7% formaldehyde (Sigma-Aldrich, ♯252549) in 1X PBS, 5 mM $MgCl_2$ (Merck, ♯1.05833.0250), 0.5 mM EGTA (Fluka, ♯03777) at room temperature for 25 min, except when using anti-DVGluT-antibody for which brains were fixed using Bouin's solution for 5 min (picric acid/formaldehyde/glacial acetic acid in proportion 15/5/1, *Daniels et al., 2008*). Brains were stained according to previously described protocols (*Sprecher et al., 2011*) and mounted in DAPI-containing Vectashield (Vector laboratories, ♯H-1200). The following primary antibodies were used: rabbit anti-GFP (Molecular Probes, ♯A6455) and sheep anti-GFP (Serotec, ♯4745–1051) (both 1:1000), mouse anti-ChAT (DHSB, ♯4B1, 1:20 for neuropil marker and 1:5 for cell neurotransmitter identification), rabbit anti-serotonin (1:1000, Sigma, S5545), rabbit anti-DVGluT (1:5000, kind gift from A. DiAntonio [*Daniels et al., 2008*]) and rabbit anti-PER (1:1000, kind gift from R. Stanewsky (*Gentile et al., 2013*). For double ChAT/DVGluT staining brains were fixed following DVGluT protocol and mouse anti-ChAT was used at 1:2. The following secondary antibodies were used: donkey anti-sheep IgG Alexa Fluor 488 (♯A11015), goat anti-rabbit IgG Alexa 488 (♯A11008) and 647 (♯21244), goat anti-mouse IgG Alexa 488 (♯A11029), 647 (♯A21235) and 568 (♯A11031) (all 1:200, Molecular Probes). Images were recorded using a Leica SP5 confocal microscope with a 63 × 1.4 NA glycerol immersion objective and LAS software. Z-projections were made in Fiji (Software, NIH) and brightness adjustment with Adobe Photoshop.

## Acknowledgements

We would like to sincerely thank Tim-Henning Humberg and our colleagues of the Sprecher lab, Akira Fushiki and Quan Yuan for fruitful discussions throughout the project. We thank the Developmental Studies Hybridoma Bank (DHSB), the Bloomington Drosophila Stock Center (BDSC), Aaron DiAntonio, Ralf Stanewsky and Boris Egger for providing fly lines and antibodies; Gerry Rubin for sharing GAL4 lines prior to publication. This work was supported by the Swiss National Science Foundation (CRSII3_136307 and 31003A_169993) and the European Research Council (ERC-2012-StG 309832-PhotoNaviNet) to S.G.S.

## Additional information

### Funding

| Funder | Grant reference number | Author |
| --- | --- | --- |
| Bundesbehörden der Schweizerischen Eidgenossenschaft | 31003A_169993 | Simon G Sprecher |
| Seventh Framework Programme | ERC-2012-StG 309832-PhotoNaviNet | Simon G Sprecher |
| Howard Hughes Medical Institute | | James W Truman<br>Marta Zlatic<br>Albert Cardona |

The funders had no role in study design, data collection and interpretation, or the decision to submit the work for publication.

### Author contributions

Ivan Larderet, Pauline MJ Fritsch, Conceptualization, Data curation, Formal analysis, Investigation, Visualization, Methodology, Writing—original draft, Writing—review and editing; Nanae Gendre,

Data curation, Visualization, Methodology; G Larisa Neagu-Maier, Data curation, Formal analysis, Visualization, Methodology, Writing—review and editing; Richard D Fetter, Casey M Schneider-Mizell, Resources, Data curation, Methodology; James W Truman, Conceptualization, Resources, Formal analysis, Investigation, Visualization, Writing—review and editing; Marta Zlatic, Resources, Data curation, Investigation; Albert Cardona, Conceptualization, Resources, Data curation, Software, Investigation, Visualization, Writing—review and editing; Simon G Sprecher, Conceptualization, Supervision, Funding acquisition, Investigation, Visualization, Methodology, Writing—original draft, Project administration, Writing—review and editing

### Author ORCIDs
Casey M Schneider-Mizell http://orcid.org/0000-0001-9477-3853
Albert Cardona http://orcid.org/0000-0003-4941-6536
Simon G Sprecher http://orcid.org/0000-0001-9060-3750

### Decision letter and Author response
Decision letter https://doi.org/10.7554/eLife.28387.020
Author response https://doi.org/10.7554/eLife.28387.021

## Additional files

**Supplementary files**
• Transparent reporting form
DOI: https://doi.org/10.7554/eLife.28387.019

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
