## [Decision Letter]

Thank you for submitting your article "Organization of the *Drosophila* larval visual circuit" for consideration by *eLife*. Your article has been reviewed by two peer reviewers, and the evaluation has been overseen by Mani Ramaswami as Reviewing Editor and K Vijay Raghavan as the Senior Editor. The reviewers have opted to remain anonymous.

The reviewers have discussed the reviews with one another and the Reviewing Editor has drafted this decision to help you prepare a revised submission.

Summary:

Larderet et al., present data that show for the first time the connectome of the early visual neurons in larval *Drosophila*. Moreover, beyond the connectome, they also use immunohistochemistry to provide the neurotransmitter types for many of the connections, which allows them to infer potential signs of connections. Obtaining this complete anatomical organization is a significant contribution that will be of great use to the field, and guide the dissection of many of the visual behaviors in fly larvae.

While the description of the visual circuits in larvae quite clear, the authors should focus because the paper in current form implies conclusions beyond their data. One joy in this sort of research is the opportunity to speculate about the circuit and its functional properties, and while this joy will not be denied, the paper should be revised to clearly separate the anatomical data from the functional speculation. Emblematic of this is the naming of one neuron ("phasic VPN"), whose name appears based on entirely speculative, inferred and unmeasured functional properties. To address this, the paper needs to be revised and rewritten in a form that makes clears distinction between the solid anatomy and the more speculative hypotheses regarding implied neuronal and network functions.

Essential revisions:

While the authors can choose how best to address the list of criticisms offered here, a strong recommendation is to have a revised paper organization and figures to more clearly distinguish the connectome and expression data from the speculation about functional properties (e.g., discuss the facts only in a first-half Results section, then talk about functional implications based on assumptions about synapse signs either in the Discussion which may be more appropriate, or in a separate second-half Results section if the authors feel this is justified).

In general, all assertions about circuit function that are not supported by other behavioral/physiological data should be discussed under "plausible or hypothesized functional interpretation of circuit architecture". Doing so will not reduce the importance of the work. Comparing larval visual circuits to adult circuits and vertebrates is even more tenuous. Therefore, even more caution is necessary. Given that the non-larval visual system perform substantial image processing, comparisons should be limited to saying that some of the features of these higher visual systems might be found in the larvae.

1) Novel neuron naming (or neuron renaming) should be based on anatomy alone, following the example adult Dmel's visual system. We note that this is mostly already done with the phasic exception.

2) Authors interpret glutamatergic synapses as "putatively inhibitory". Glutamatergic synapses can be either excitatory or inhibitory, and without other evidence it is not appropriate to suggest that they are inhibitory. Similarly, the authors predicate much of their speculation on the assumption that cholinergic neurons will be excitatory. However, there are metabotropic acetylcholine receptors in Dmel that could be inhibitory (see Shinomiya et al., (2014) for a suggestion of their use). This is an incredibly important caveat for virtually all the analysis in the paper, and it should be made more clearly. In 1H and elsewhere, one could color code the connections by transmitter, rather than using the standard symbols for excitatory/inhibitory connections. Then later, when speculating more about the circuit function, one could add in those symbols.

3) The paper hops around a lot. For instance, Figure 1 are called in the text, but then the results move right on to Figure 2, without statements about the other panels. The first result section cycles through all figures, without calling all panels, then the next result section cycles through them again, looking at different panels. In reorganizing the paper to clearly distinguish data from functional speculation, it would be useful to make sure all figure panels are called in order (and all supplementary figures).

4) Authors argue both that Rh5 and Rh6 are in parallel and for substantial cross-talk. For example, in the Discussion section they say "clear separation of visual information". Later on they say – "flow of information is convergent". It can be one or the other.

There are several points on this:

A) Isn't it already known that either Rh5 or Rh6 can mediate circadian entrainment. Does that not mean that Rh5 and Rh6 are not in parallel?

B) Another confusion here is that despite Rh5 being the major input into VPNs, VPN responses are interpreted based entirely on the indirect input from Rh6 via VLN (subsection “The Rh6-PRs-VLNs pathway may compute variations in light intensity “). The authors are assuming that because Rh5-VPN connection is not under pre-synaptic inhibition, VPN response to Rh5 simply scales with absolute light intensity. This assumption is unjustified. In general, Rh5-VPN responses have been largely ignored in the discussion.

C) The functional effect of circuitry downstream of Rh6 is over-interpreted.

5) Comparison to both *Drosophila* adults and vertebrate seem stretched. Functionally speaking, Rh6 in larva seems to subserve a circadian function. In that sense, they are similar to melanopsin-expressing cells in the vertebrates. Therefore, it would be difficult for the authors to rigorously refute that Rh5= rods and cones; rh6=melanopsin. Similarly, functionally, the prominent role of r1-6 in motion vision is different from Rh6's role in the larvae. Also, isn't the more appropriate comparison is that between the larval circuit and the Hofbauer-Buchner eyelets. Other alternative interpretations of the observations are possible (see minor points 19 and 22 for example). In general, comparison with vertebrate or adults should be moderated by discussing differences, caveats and problems.

6) Speculation that Rh5 neuron circuits encode absolute intensity is surprising, since phototransduction and various feedbacks in most PRs, from vertebrates to invertebrates, appear to be designed to adapt to the mean intensity of incident light. Is there evidence this is not true to larval PRs? In the adult visual system, LMC neurons achieve a strong derivative of their inputs, apparently without much lateral interactions – couldn't this be happening where Rh5 PRs synapse onto a single postsynaptic VPN target?

7) Inputs into the LONp are not fully clear. The summary diagram of Figure 1 should include LONp and its components – Phasic VPN and Tiny VLN. So should Figure 6.

---

## [Author Response]

Essential revisions:

*While the authors can choose how best to address the list of criticisms offered here, a strong recommendation is to have a revised paper organization and figures to more clearly distinguish the connectome and expression data from the speculation about functional properties (e.g., discuss the facts only in a first-half Results section, then talk about functional implications based on assumptions about synapse signs either in the Discussion which may be more appropriate, or in a separate second-half Results section if the authors feel this is justified).*

This is indeed a valid point. We agree that a clear separation from the actual result section and interpretation will be adequate. Accordingly, we removed all interpretation of interaction within the network and potential function from the Results and Discussion section (see below essential point 2 and minor point 10). As suggested, we have included this now in the Discussion section. Moreover, we removed all interpretation of excitatory or inhibitory inputs in the figure legends (Figure 3).

*In general, all assertions about circuit function that are not supported by other behavioral/ physiological data should be discussed under "plausible or hypothesized functional interpretation of circuit architecture". Doing so will not reduce the importance of the work. Comparing larval visual circuits to adult circuits and vertebrates is even more tenuous. Therefore, even more caution is necessary. Given that the non-larval visual system perform substantial image processing, comparisons should be limited to saying that some of the features of these higher visual systems might be found in the larvae.*

As suggested we have now included all aspects concerning circuit function in a new section in the discussion entitled “Hypothesized functional interpretation of circuit architecture”.

*1) Novel neuron naming (or neuron renaming) should be based on anatomy alone, following the example adult Dmel's visual system. We note that this is mostly already done with the phasic exception.*

It is indeed true that currently there is no functional evidence for a phasic function of the “Phasic VPN”. We have accordingly renamed this third-order neuron to “visual projection-local interneuron: VPLN”, a name that clearly highlights its anatomical features of having both significant local connections and axonal projections.

*2) Authors interpret glutamatergic synapses as "putatively inhibitory". Glutamatergic synapses can be either excitatory or inhibitory, and without other evidence it is not appropriate to suggest that they are inhibitory. Similarly, the authors predicate much of their speculation on the assumption that cholinergic neurons will be excitatory. However, there are metabotropic acetylcholine receptors in Dmel that could be inhibitory (see Shinomiya et al., (2014) for a suggestion of their use). This is an incredibly important caveat for virtually all the analysis in the paper, and it should be made more clearly. In 1H and elsewhere, one could color code the connections by transmitter, rather than using the standard symbols for excitatory/inhibitory connections. Then later, when speculating more about the circuit function, one could add in those symbols.*

The points raised here are indeed valid. Accordingly we now uniquely refer to the presence of neurotransmitters, without any interpretation of their putative function, in the results section. For putative excitatory and inhibitory functions we included a paragraph in the discussion to better introduce our assumptions (in the Rh6-PRs-VLNs pathway part of the discussion). We also rephrased relevant sections of the discussion to clarify that these are hypothesized functions. Finally, in the figures we removed the standard symbols for excitatory/inhibitory connections to use a color code for neurotransmitter types: black arrows for cholinergic, red arrows for glutamatergic, brown arrows for other types; in Figure 2 (previous Figure 1), in the whole Figure 3 (previous Figure 2 and Figure 3) and Figure 3—figure supplement 1 (previous Figure 3—figure supplement 3). The symbols for “inhibition” and “excitation” are only used in the complete model of the connectome provided in the supplement 2 of Figure 3, which is only referred to in the discussion.

*3) The paper hops around a lot. For instance, Figure 1 are called in the text, but then the results move right on to Figure 2, without statements about the other panels. The first result section cycles through all figures, without calling all panels, then the next result section cycles through them again, looking at different panels. In reorganizing the paper to clearly distinguish data from functional speculation, it would be useful to make sure all figure panels are called in order (and all supplementary figures).*

We thank the reviewers for this suggestion. We now reorganized the manuscript and figures in a fashion that the figures order closely follows the text. The following changes were made:

Figure 1 now presents all the important interneurons of the LON. It integrates panels A and B from the previous Figure 1, panels B and C from the previous Figure 2, and panels E to L from the previous Figure 3 except the panel I of the previous Figure 3 that was removed.

Figure 2 now focuses on the structural organization of the LON and presents the simplified connectome. It integrates the panels C to H of the previous Figure 1.

Figure 3 now focuses on the detailed connections of the important LON interneurons. It integrates the panel A from the previous Figure 2 and the panels A to D from the previous Figure 3.

Figure 4, Figure 5 and Figure 6 stayed the same.

Figure 1—figure supplement 1 (animation movie) stayed the same.

Figure 1—figure supplement 2 now integrates the neurotransmitters identification of all the main interneurons of the LON. It fuses all the panels from the previous Figure 2—figure supplement 1 and Figure 3—figure supplement 1.

Figure 1—figure supplement 3 is the previous Figure 1—figure supplement 4.

Figure 1—figure supplement 4 is the previous Figure 3—figure supplement 2.

Figure 1—figure supplement 5 (animation movie) is the previous Figure 3—figure supplement 4.

Figure 2—figure supplements 1 and 2 (excel tables) are the previous Figure 1—figure supplement 2 and Figure 1—figure supplement 3.

Figure 3—figure supplement 1 is the previous Figure 3—figure supplement 3.

Figure 3—figure supplement 2 is the previous Figure 1—figure supplement 5.

Figure 4—figure supplement 1 and Figure 5—figure supplement 1 stayed the same.

*4) Authors argue both that Rh5 and Rh6 are in parallel and for substantial cross-talk. For example, in the Discussion section they say "clear separation of visual information". Later on they say "flow of information is convergent". It can be one or the other.*

We fully agree and apologize that this has not been clearly defined before. While the two PRs provide “parallel” outputs onto the two sets of visual interneurons, the main feature of the entire pathway in the LON is the convergence of Rh6- and Rh5-input. We have therefore removed the term “parallel” in the Abstract and in the title of the corresponding Results section. Moreover, in subsection “The visual circuit supports previous behavioral observe “we clarified that the separation of visual information is occurring only at the first synapse level and then converge onto the VPN.

*There are several points on this:*

*A) Isn't it already known that either Rh5 or Rh6 can mediate circadian entrainment. Does that not mean that Rh5 and Rh6 are not in parallel?*

Yes, this is indeed correct and in agreement with the capacity of both PR subtypes to entrain the larval circadian rhythm, we found that the four Pdf-LaNs receive inputs directly from both PRs. We now highlight the fact that the Pdf-LaNs are therefore an exception to the segregation of PRs inputs at the first synaptic level, subsection “Central brain feedback via octopaminergic/tyraminergic and serotonergic neurons”(and by removing “parallel” as stated above).

*B) Another confusion here is that despite Rh5 being the major input into VPNs, VPN responses are interpreted based entirely on the indirect input from Rh6 via VLN (subsection “The Rh6-PRs-VLNs pathway may compute variations in light intensity “). The authors are assuming that because Rh5-VPN connection is not under pre-synaptic inhibition, VPN response to Rh5 simply scales with absolute light intensity. This assumption is unjustified. In general, Rh5-VPN responses have been largely ignored in the discussion.*

There is indeed no physiological data available on the light response properties of larval PRs and thus indeed the assumption of Rh5-PRs providing absolute light intensity is hypothetical. We have therefore removed this from the Results section and added a first section about these PRs in the discussion (see also point 6).

*C) The functional effect of circuitry downstream of Rh6 is over-interpreted.*

We apologize for the over-interpretation and as stated above we have moved all hypothetical aspects into the Discussion section. We also discuss alternative possibilities.

*5) Comparison to both Drosophila adults and vertebrate seem stretched. Functionally speaking, Rh6 in larva seems to subserve a circadian function. In that sense, they are similar to melanopsin-expressing cells in the vertebrates. Therefore, it would be difficult for the authors to rigorously refute that Rh5= rods and cones; rh6=melanopsin. Similarly, functionally, the prominent role of r1-6 in motion vision is different from Rh6's role in the larvae. Also, isn't the more appropriate comparison is that between the larval circuit and the Hofbauer-Buchner eyelets. Other alternative interpretations of the observations are possible (see minor points 19 and 22 for example). In general, comparison with vertebrate or adults should be moderated by discussing differences, caveats and problems.*

We thank the reviewers for raising this point. According to the suggestion we have rephrased this Discussion section highlighting the comparison here as one possible interpretation and also commenting on other possible hypotheses. Moreover, we have adapted the title of the Figure 6 to clarify that we are comparing structural features that are clear in our view and putative functions that we propose. Since both Rh5-PRs and Rh6-PRs are individually sufficient for circadian clock-entrainment, we did not include a comparison with melanopsin neurons.

*6) Speculation that Rh5 neuron circuits encode absolute intensity is surprising, since phototransduction and various feedbacks in most PRs, from vertebrates to invertebrates, appear to be designed to adapt to the mean intensity of incident light. Is there evidence this is not true to larval PRs? In the adult visual system, LMC neurons achieve a strong derivative of their inputs, apparently without much lateral interactions – couldn't this be happening where Rh5 PRs synapse onto a single postsynaptic VPN target?*

It is indeed true that similar mechanisms of light adaptation may also be present in the larval PRs. Since electrophysiological data from larval PRs are lacking, neither possibility can be excluded. It is well possible (or even likely) that larval PRs show comparable features to adult PRs. We have therefore modified the discussion referring to both possibilities in a new section at the beginning of the hypothetical functional interpretation part of the discussion. Moreover, we replaced the term “absolute” by “ambient” everywhere in the text.

*7) Inputs into the LONp are not fully clear. The summary diagram of Figure 1 should include LONp and its components – Phasic VPN and Tiny VLN. So should Figure 6.*

We adapted the figures accordingly to also include the LONp (thus the corresponding third order neurons) in Figure 2 and Figure 6, as well as in the Figure 3—figure supplement 2.